# Indole-3-Carbinol Inhibits the Growth of Endometriotic Lesions by Suppression of Microvascular Network Formation

**DOI:** 10.3390/nu14224940

**Published:** 2022-11-21

**Authors:** Jeannette Rudzitis-Auth, Madeleine Becker, Claudia Scheuer, Michael D. Menger, Matthias W. Laschke

**Affiliations:** Institute for Clinical and Experimental Surgery, Saarland University, 66421 Homburg/Saar, Germany

**Keywords:** endometriosis, indole-3-carbinol, angiogenesis, vascularization, proliferation, mouse model, high-resolution ultrasound imaging

## Abstract

Endometriosis represents an estrogen-dependent disorder with a complex pathophysiology. Phytochemicals are promising candidates for endometriosis therapy, because they simultaneously target different cellular processes involved in the pathogenesis of endometriosis. Herein, we analyzed whether indole-3-carbinol (I3C) suppresses the development of endometriotic lesions, which were surgically induced by fixation of uterine tissue samples (diameter: 2 mm) from female BALB/c donor mice to the peritoneum of recipient animals. The mice received either I3C or vehicle (control) by peroral administration once per day. Growth, cyst formation, cell proliferation, microvascularization and protein expression of the lesions were assessed by high-resolution ultrasound imaging, caliper measurements, histology, immunohistochemistry and Western blotting. I3C inhibited the vascularization and growth of endometriotic lesions without inducing anti-angiogenic and anti-proliferative side effects on reproductive organs. This was associated with a significantly reduced number of proliferating stromal and endothelial cells and a lower expression of the pro-angiogenic signaling molecules vascular endothelial growth factor receptor-2 (VEGFR2), phosphoinositide 3-kinase (PI3K) and phosphorylated extracellular signal-regulated kinase (pERK) within I3C-treated lesions when compared to controls. These findings indicate that I3C effectively inhibits endometriotic lesion formation in mice. Thus, further studies should clarify whether I3C may be also beneficial for the prevention and therapy of the human disease.

## 1. Introduction

Endometriosis represents a benign, estrogen-dependent disorder, which affects ~5–10% of mainly premenopausal women [1,2]. This disease is defined by the appearance of ectopic stromal and glandular endometrial tissue outside the uterine cavity and is associated with a variety of clinical symptoms, such as dysmenorrhea and chronic abdominal or pelvic pain, as well as infertility [3]. This, in turn, markedly impairs the quality of life of affected patients and causes considerable economic costs in the health care system [4].

The management of endometriosis includes surgical and pharmacological approaches. While the surgical removal of endometriotic lesions is primarily indicated for the treatment of severe or resistant cases, drug therapy is the first choice to treat pain symptoms and to prevent the progression of the disease or its recurrence after surgery [5]. For this purpose, hormonal suppression of endometriotic tissue is achieved by gonadotropin-releasing hormone (GnRH) agonists, aromatase inhibitors, oral contraceptives, selective progesterone receptor modulators and danazol [6]. However, these compounds bear the risk of side effects, such as irregular bleeding, postmenopausal symptoms and osteoporosis [7]. Hence, the establishment of novel treatment concepts, which are highly efficient and only induce minor side effects, is a major subject of current endometriosis research.

The pathophysiology of endometriosis is complex [8]. It involves the development of new blood vessels, also referred to as angiogenesis, which is essential to guarantee the survival of endometriotic tissue at ectopic sites [9,10]. Moreover, it is characterized by the activation of the immune system and the recruitment of inflammatory cell types into endometriotic lesions, resulting in enhanced tissue proliferation [11,12]. Targeting these processes simultaneously by phytochemicals with a pleiotropic action profile represents a promising strategy to suppress the growth of ectopic endometrium without severe side effects [13]. This view is supported by many studies demonstrating that the application of phytochemicals, such as epigallocatechin-3-gallate, curcumin, puerarin, 4-hydroxybenzyl alcohol, resveratrol and xanthohumol, is capable of inducing the regression of endometriotic lesions [14,15,16,17,18,19,20,21].

A potential phytochemical for the therapy of endometriosis is indole-3-carbinol (I3C). This hydrolysis product of glucobrassicin is a component of cruciferous vegetables, such as broccoli, cauliflower, kohlrabi and radish [22]. I3C inhibits the formation of newly developing blood vessels by decreasing the secretion of vascular endothelial growth factor (VEGF) and the activity of matrix metalloproteinase (MMP)-2 and -9 [23,24]. Moreover, I3C suppresses estrogen receptor (ER)α signaling in cell lines of breast and cervical cancer [25,26].

In line with these findings, we investigated in this study the effect of I3C on surgically induced endometriotic lesions in mice. For this purpose, uterine tissue samples from donor mice were transplanted to the abdominal wall of I3C- and vehicle-treated recipient mice to induce endometriotic lesions. The formation and cellular signaling of these lesions were investigated by high-resolution ultrasound imaging, histology, immunohistochemistry and Western blotting.

## 2. Materials and Methods

### 2.1. Animals

Female BALB/c mice (age: 12–20 weeks; body weight: 18–25 g; Institute for Clinical and Experimental Surgery, Saarland University, Homburg/Saar, Germany) were used for this study. Up to six mice were kept in one cage under a 12-h day/night cycle with free access to tap water and standard pellet food (Altromin, Lage, Germany).

All animal experiments were performed according to the German legislation on protection of animals and the National Institutes of Health Guide for the Care and Use of Laboratory Animals (Institute of Laboratory Animal Resources, National Research Council, Washington, DC, USA) and were approved by the local governmental animal protection committee (permission numbers: 47/2016; 11/2017).

### 2.2. Vaginal Lavage

To ensure comparable steroid hormone levels of individual mice, the estrous cycle stage was assessed by vaginal lavage prior to the surgical induction of endometriotic lesions. This was achieved by pipetting 15 µL of 0.9% saline solution into the vagina of the mice and by the analysis of the gained cell suspension under a phase contrast microscope (CH-2; Olympus, Hamburg, Germany). Only animals in the stage of estrus served as donors for the harvesting of uterine tissue samples and recipients for the surgical induction of endometriotic lesions.

### 2.3. Isolation of Uterine Tissue Samples

To isolate uterine tissue samples, donor BALB/c mice in the stage of estrus were anesthetized by an intraperitoneal injection of 75 mg/kg ketamine (Ursotamin^®^; Serumwerke Bernburg, Bernburg, Germany) and 15 mg/kg xylazine (Rompun^®^; Bayer, Leverkusen, Germany). After midline laparotomy, the uterine horns of the animals were excised and transferred into a culture dish filled with Dulbecco’s modified Eagle medium (DMEM; PAN Biotech, Aidenbach, Germany; 10% fetal calf serum, 100 U/mL penicillin, 0.1 mg/mL streptomycin (Thermo Fisher Scientific, Dreieich, Germany)). Thereafter, they were opened longitudinally and 2 mm tissue samples were carefully removed with a biopsy punch (Stiefel Laboratorium GmbH, Offenbach am Main, Germany) (Figure 1A).

### 2.4. Induction of Endometriotic Lesions

To study the effects of I3C on peritoneal endometriotic lesions, uterine tissue samples were transplanted into the abdomen of BALB/c mice in the stage of estrus [27]. The animals were anesthetized by an intraperitoneal injection of ketamine (75 mg/kg body weight; Ursotamin^®^, Serumwerke Bernburg) and xylazine (15 mg/kg body weight; Rompun^®^, Bayer). After midline laparotomy, two (for ultrasound analyses) or three tissue samples (for Western blot analyses) were sutured with 6–0 Prolene (Ethicon Products, Norderstedt, Germany) to the peritoneum at each side of the abdominal wall (Figure 1B). Thereafter, the abdomen was closed again with running 5–0 Prolene muscle and skin sutures.

### 2.5. High-Resolution Ultrasound Imaging and Analysis

By means of a real-time microvisualization (RMV^TM^) 704 Scanhead (VisualSonics, Toronto, ON, Canada) with a center frequency of 40 MHz and a focal depth of 6 mm, the endometriotic lesions were analyzed with a Vevo 770^TM^ high-resolution ultrasound imaging system (VisualSonics) [28]. Under 2% isoflurane anesthesia, the mice were fixed in supine position on a heated stage and the fur of the abdomen was removed by chemical depilation (Nair hair removal lotion; Church & Dwight Canada Corp., Mississauga, ON, Canada). To analyze the ultrasound images, a three-dimensional reconstruction and analysis software from VisualSonics (Vevo 770 V2.3.0) was used. The borders of the lesions and their cyst-like dilated endometrial glands (in the following referred to as cysts) were marked in parallel sections with a step size of 200 µm, and the total volume of the endometriotic lesions and the volume of their stromal tissue and cysts (in mm^3^) were calculated by manual image segmentation [29]. Furthermore, the growth of the lesions and their stromal tissue (as percentage of the initial lesion and stromal tissue volume) as well as the number of cyst-containing lesions (as percentage) were calculated.

After the last ultrasound imaging, the abdomens of the mice were opened under a stereomicroscope. The largest diameter (D1) and the perpendicularly aligned diameter (D2) of the lesions were assessed with a digital caliper (Figure 1C) to calculate the lesion size (S) with the formula S = D1 × D2 × π/4 [30]. Subsequently, the lesions as well as the ovaries and uterine horns of the recipient mice were excised and further processed for histology and immunohistochemistry.

### 2.6. Histology and Immunohistochemistry

The endometriotic lesions, ovaries and uterine horns were fixed in formalin and embedded in paraffin. Using a standard protocol, sections of 3 µm thickness were stained with hematoxylin and eosin (HE).

Proliferating cells within the samples were detected by a rabbit polyclonal antibody against the proliferation marker Ki67 (1:500; Abcam, Cambridge, UK) followed by a goat anti-rabbit biotinylated secondary antibody (ready-to-use, Abcam) and avidin-peroxidase (1:50, Sigma-Aldrich, St. Louis, MO, USA). 3-Amino-9-ethylcarbazole (AEC Substrate System, Abcam) served as chromogen, and counterstaining was achieved with hemalaun. The number of Ki67^+^ cells in four regions of interest within the tissue were counted to calculate the fraction of proliferating cells (%).

Microvessels within the samples were detected by a monoclonal rat anti-mouse antibody against the endothelial cell marker CD31 (1:100; Dianova GmbH, Hamburg, Germany), followed by a goat anti-rat IgG Alexa555 secondary antibody (1:100; Invitrogen, Darmstadt, Germany). Nuclear staining was performed with Hoechst 33342 (2 µg/mL; Sigma-Aldrich). The microvessel density (mm^−2^) was assessed under a BZ-8000 microscope (Keyence, Osaka, Japan) by counting the overall number of CD31^+^ microvessels, which was then divided by the analyzed area of stromal tissue.

Proliferating Ki67^+^/CD31^+^ endothelial cells within the endometriotic lesions were detected by a monoclonal rat anti-mouse antibody against CD31 (1:100; Dianova GmbH), followed by a goat anti-rat IgG Alexa488 secondary antibody (1:200; Invitrogen) and by a rabbit polyclonal antibody against Ki67 (1:200; Abcam), followed by a goat anti-rabbit IgG Alexa555 secondary antibody (1:50; Invitrogen). Nuclear staining was performed with Hoechst 33,342 (2 µg/mL; Sigma-Aldrich). The fraction of Ki67^+^/CD31^+^ endothelial cells (%) was assessed under a BX60 microscope (Olympus).

### 2.7. Western Blot Analysis

Endometriotic lesions treated with I3C or vehicle were lysed on ice for 1 min by mechanical homogenization (Miccra, D-1; Miccra GmbH, Heitersheim, Deutschland) in lysis buffer, which contained 10 mM Tris (pH 7.5), 10 mM NaCl, 0.1 mM EDTA, 0.5% Triton-X 100, 0.02% NaN_3_, phenylmethylsulfonyl fluoride (1:500 *v*/*v*), protease inhibitor cocktail (1:100 *v*/*v*; Sigma-Aldrich) and phosphatase inhibitor cocktail (1:100 *v*/*v*; Sigma-Aldrich). The lysate was centrifuged for 30 min at 21,000× *g* (4°C) and the supernatant saved as whole protein fraction. After determination of the protein concentrations using the Lowry method with bovine serum albumin standard, 20 µg protein were added per lane on 10% sodium dodecyl sulfate (SDS) polyacrylamide gels and transferred to a polyvinylidene difluoride (PVDF) membrane (BioRad, München, Deutschland). After blocking the non-specific binding sites with Blotting-Grade Blocker (BioRad), the membranes were incubated overnight at 4 °C with a monoclonal rabbit anti-vascular endothelial growth factor receptor-2 (VEGFR2) antibody (1:100; Cell Signaling Technology, Frankfurt, Germany), a polyclonal rabbit anti-VEGF antibody (1:50; Abcam), a polyclonal rabbit anti-extracellular signal-regulated kinase (ERK) antibody (1:300; Abcam), a monoclonal mouse anti-phosphorylated (p)ERK antibody (1:300; Abcam), a monoclonal rabbit anti-protein kinase B (AKT) antibody (1:300; Cell Signaling Technology), a monoclonal rabbit anti-pAKT1/2/3 antibody (1:100; Cell Signaling Technology), a polyclonal rabbit anti-cyclooxygenase (COX)-2 antibody (1:300; Abcam), a monoclonal mouse anti-phosphoinositide 3-kinase (PI3K) antibody (1:50; Santa Cruz Biotechnology, Heidelberg, Deutschland), a monoclonal mouse anti-nuclear factor kappa-light-chain-enhancer of activated B cells (NF-κB) antibody (1:50, R&D Systems, Wiesbaden, Germany), a monoclonal rabbit anti-ERα antibody (1:50, Cell Signaling), a polyclonal rabbit anti-ERβ antibody (1:50, Abcam) or a mouse monoclonal anti-β-actin antibody (1:5000; Sigma Aldrich). This was followed by the corresponding anti-mouse IgG (1:1500; Dako/Agilent, Hamburg, Germany) or anti-rabbit IgG secondary antibodies conjugated to horseradish peroxidase (HRP) (1:1000; R&D Systems). For visualization of the protein expression, Clarity Western ECL substrate (BioRad) was used. Images were generated by a Chemocam device (Intas, Science Imaging Instruments, Göttingen, Germany). The intensity of individual bands was measured with the Lab Imaged 1D software (Intas, Science Imaging Instruments).

### 2.8. Experimental Protocol

In a first set of experiments, 64 uterine tissue samples from four donor mice were transplanted to the left and the right abdominal wall of 16 recipient mice. The animals received either 120 mg/kg I3C (Sigma-Aldrich) (*n* = 8) or 100 µL corn oil (Sigma-Aldrich) as vehicle (control; *n* = 8) by daily intragastric injection for 28 days. The newly developing endometriotic lesions were analyzed by high-resolution ultrasound imaging directly after tissue transplantation (d0), as well as on days 7, 14, 21 and 28. In addition, the lesion size was measured with a digital caliper at the end of the in vivo experiments. Thereafter, the lesions, uterine horns and ovaries were processed for histology and immunohistochemistry.

In a second set of experiments, a total of 48 uterine tissue samples from four donor mice were transplanted into 12 recipient mice, which were treated daily with either 120 mg/kg I3C (*n* = 6) or 0.1 mL corn oil (*n* = 6) as vehicle by intragastric injection for 7 days. Thereafter, the lesions, uterine horns and ovaries were processed for histology and immunohistochemistry.

In a third set of experiments, a total of 36 uterine tissue samples from four donor mice were transplanted into the abdominal cavity of six recipient animals, which were treated daily with either 120 mg/kg I3C (*n* = 3) or 0.1 mL corn oil as vehicle (control; *n* = 3) by intragastric injection for seven days. Thereafter, the tissue samples were processed for Western blot analyses.

### 2.9. Statistics

For the experiments, groups of equal size using randomization were used. The analyses were performed in a blinded manner. Data were first analyzed for normal distribution and equal variance. In case of parametric data, differences between the two experimental groups (I3C vs. control) were assessed by the unpaired Student’s t-test. In the case of non-parametric data, differences between the two experimental groups were assessed by the Mann–Whitney rank sum test. All data are given as mean ± standard error of the mean (SEM). Statistical significance was accepted for *p* < 0.05.

## 3. Results

### 3.1. Effect of I3C on the Development of Endometriotic Lesions

In a first set of experiments, we investigated the development of surgically induced endometriotic lesions by means of repeated high-resolution ultrasound imaging (Figure 2A–H). At the beginning of this analysis, the transplanted uterine tissue samples in both I3C- and vehicle-treated mice exhibited an equal volume, indicating standardized baseline conditions (Figure 2C). Throughout the following 28-day observation period, the uterine grafts of the control group progressively grew and developed into typical murine endometriotic lesions with cyst-like dilated glands surrounded by endometrial stroma (Figure 2A,C,D). In contrast, the developing lesions in I3C-treated mice did not markedly grow over time and, thus, presented with a significantly lower lesion volume and growth rate on day 28 when compared to controls (Figure 2B–D). This was caused by the suppression of stromal tissue growth (Figure 2E,F) and reduced cyst volumes (Figure 2G). However, the fraction of cyst-containing lesions was comparable in both groups (Figure 2H). In line with these results, the lesions revealed a significantly smaller lesion size in I3C-treated mice on day 28 when compared to controls, as assessed by caliper measurements (Figure 3A). Additional histological analyses confirmed, in both groups, the development of endometriotic lesions containing endometrial stroma cells and glandular epithelial cells (Figure 3B–E).

### 3.2. Effect of I3C on Proliferation and Angiogenesis within Endometriotic Lesions

To analyze the effect of I3C on the cellular proliferation within developing endometriotic lesions, proliferating stromal and glandular endometriotic cells were detected by the immunohistochemical marker Ki67^+^. This analysis revealed a significantly reduced number of proliferating endometrial stromal cells in lesions of I3C-treated animals on day 7 when compared to controls (Figure 4A,B,M). In contrast, we did not detect any differences in the proliferative activity of the glandular epithelium within the lesions of both groups on days 7 or 28 (Figure 4A–D,M).

In addition, the effect of I3C on the formation of new microvascular networks in developing endometriotic lesions was investigated. For this purpose, the density of CD31^+^ microvessels was determined by immunohistochemistry. On day 7, the lesions of I3C- and vehicle-treated mice exhibited a comparable microvessel density (Figure 4E,F,N). However, at this early time point, the microvessels within lesions of I3C-treated animals contained significantly fewer proliferating Ki67^+^/CD31^+^ endothelial cells when compared to those of control mice (Figure 4I,J,O). This interesting finding may also explain the observation that the microvessel density of I3C-treated lesions was significantly lower on day 28 (Figure 4G,H,N), whereas the fraction of proliferating Ki67^+^/CD31^+^ endothelial cells did not differ anymore between the two groups at this later time point (Figure 4K,L,O).

### 3.3. Effect of I3C on the Protein Expression of Endometriotic Lesions

To gain deeper insights into the molecular mechanisms underlying the inhibitory action of I3C on the development of endometriotic lesions, we analyzed different signaling pathways and their related components. Western blot analyses of the lesions revealed that I3C strongly inhibits the expression of PI3K without affecting the phosphorylation of AKT or the expression of NF-ĸB (Figure 5A–D). Moreover, we found that the phytochemical significantly downregulates the expression of VEGFR2 without affecting the expression of VEGF (Figure 5E–G). Among others, VEGF/VEGFR2 signaling is responsible for the induction of ERK1/2 and COX-2 [31,32]. Hence, we also detected a markedly suppressed phosphorylation of ERK1/2 in endometriotic lesions of I3C-treated mice when compared to controls (Figure 5E,H). However, the expression of COX-2 was not affected by the treatment with the phytochemical (Figure 5I,J).

Finally, we analyzed the expression of ERα and ERβ. The two estrogen receptors play a major role in the pathogenesis of endometriosis and are differently expressed in eutopic and ectopic endometria [33]. Unexpectedly, we detected a significantly increased expression of ERα in endometriotic lesions of I3C-treated mice, whereas the expression of ERβ was comparable in lesions of I3C- and vehicle-treated animals (Figure 5I,K,L).

### 3.4. Effect of I3C on the Proliferation and Microvessel Density within the Repoductive Organs

Finally, we performed additional histological and immunohistochemical analyses of the ovaries and uterine horns to detect possible side effects of I3C treatment on the reproductive organs. We found that the ovaries and uterine horns of I3C- and vehicle-treated mice exhibited a comparable histomorphology (Figure 6A–D). Moreover, they did not differ in their proliferative activity, as indicated by a comparable fraction of Ki67^+^ cells within their ovarian follicles and endometrial glands as well as within their stroma (Figure 6E–H,M,N). In addition, I3C treatment did not affect the microvessel density within the ovaries and the eutopic endometrium of the uterine horns (Figure 6I–L,O).

## 4. Discussion

I3C exhibits anti-oxidative, anti-inflammatory, anti-proliferative, anti-angiogenic and pro-apoptotic properties [34]. These are beneficial for the therapy of endometriosis, which is typically associated with blood vessel formation, inflammation and tissue proliferation [35,36]. Accordingly, we could show that I3C suppresses the vascularization and growth of endometriotic lesions in a murine endometriosis model.

In our proof-of-principle study we surgically induced endometriotic lesions by grafting uterine tissue samples from donor mice into the abdominal cavity of syngeneic recipient mice. The animals were treated daily by peroral administration of 120 mg/kg I3C, which has previously been shown to effectively suppress murine leukemia [37]. This dose in mice corresponds to 9.7 mg/kg or 679 mg of I3C for a 70 kg person, according to the dose translation method by Reagan-Shaw et al. [38]. Thus, the I3C dose used in our experiments is within the dose range of 400–800 mg daily, which has already been applied in a phase-I study for the treatment of women with high risk of breast cancer [39]. Importantly, this dose range was well tolerated by the patients and did not induce severe side effects [39]. It should further be considered that I3C is rapidly absorbed after oral ingestion and distributed to well-vascularized tissues, such as the kidney, liver, lungs and brain [22]. Since I3C is rapidly converted under acidic conditions, its effect is partly mediated by the formation of active metabolites. This supports the concept that I3C acts as a pro-drug.

Several studies reported that I3C inhibits the proliferation of different tumor cell lines in vitro and in vivo [40,41,42,43]. In line with this, we also counted a lower number of proliferating endometrial stromal cells within I3C-treated endometriotic lesions when compared to controls. Accordingly, our repeated ultrasound analyses revealed that I3C-treated lesions do not markedly grow during a period of 28 days, and even show a regression of their stromal tissue over time. Although not further analyzed in our study, the latter observation may be partly explained by the pro-apoptotic activity of I3C [34]. Another explanation may be the fact that the survival of endometriotic lesions is dependent on a sufficient vascularization to guarantee their supply of oxygen and essential nutrients [35,44]. Our immunohistochemical analyses showed that this vascularization was impaired in I3C-treated animals, as indicated by a significantly reduced microvessel density on day 28 when compared to controls. The additional analysis of lesions on day 7 revealed that this resulted from a direct anti-proliferative effect of I3C on endothelial cells within newly developing blood vessels during the early phase of lesion development, which is characterized by a high angiogenic activity of the ectopic endometrial tissue [44]. This view is supported by previous studies demonstrating that I3C inhibits the proliferation of endothelial cells and their ability to form tube-like structures by suppressing intra- and extracellular angiogenesis-driving signaling molecules, such as ERK1/2, nitric oxide, VEGF and MMP-9 [40,45,46,47].

To gain deeper insights in the molecular mechanisms underlying the inhibitory action of I3C on the development of endometriotic lesions, we additionally performed Western blot analyses. Eutopic and ectopic endometrial tissue of endometriosis patients exhibits an increased expression of PI3K and its major downstream target pAKT/AKT when compared to endometrium from healthy controls [48]. Interestingly, we detected a significantly reduced expression of PI3K in I3C-treated lesions. This may explain the anti-proliferative effects of our phytochemical treatment, because PI3K is a central regulator of cell proliferation in response to intra- and extracellular signals [49]. However, we did not detect a reduced expression of pAKT/AKT. This unexpected result is in line with an in vitro study in MCF-7 breast cancer cells, in which pAKT/AKT expression was also not affected by I3C treatment [50]. Moreover, it supports the concept that AKT can be regulated by PI3K-independent pathways, such as p38 [51,52]. Additionally, it is well known from cancer studies that PI3K can exert its anti-proliferative effects via AKT-independent signaling [53]. Our result indicates that this may have also been the case in our study, and suggests that additional investigations are required to unravel the complex relationship between PI3K and AKT in endometriosis.

Our Western blot analyses further revealed a markedly reduced expression of VEGFR2 and its downstream target pERK/ERK in I3C-treated endometriotic lesions. This finding shows that I3C is a potent inhibitor of this signaling pathway, which has previously been reported to regulate angiogenesis in endometriosis [18,54]. In this context, it should be noted that ERK1/2 can stimulate the expression of COX-2, which promotes the development of new microvessels in developing endometriotic lesions [44,55]. However, in our study, COX-2 expression levels remained unaffected in I3C-treated lesions. This finding may be due to the fact that several other pathways, such as NF-kB and AKT signaling, which herein were not altered by I3C treatment, are also involved in the regulation of COX-2 expression in endometriosis [56].

In addition, we detected a significantly higher expression of ERα in I3C-treated lesions when compared to controls. A similar observation was made by Nouri Emamzadeh et al. [57], reporting that I3C strongly stimulates the re-expression of ERα in triple negative breast cancer cells. At first glance, this I3C effect seems to be counterproductive for the treatment of endometriotic lesions, because endometriosis is an estrogen-dependent disease [58]. However, it should be noted that I3C targets estrogen-dependent mechanisms on various levels. In fact, I3C is known to act as an inhibitor of ERα-responsive gene expression [26]. Moreover, it induces the expression of cytochrome P450 1A1, which converts estrone to 2-hydroxyestrone, a metabolite with anti-proliferative, anti-angiogenic and pro-apoptotic properties [59,60]. Therefore, it is plausible that I3C suppressed the development of endometriotic lesions in our study despite its clear stimulatory effect on ERα expression.

Finally, we also analyzed the uterine horns and ovaries of I3C- and vehicle-treated mice. We found that the histomorphology, vascularization and proliferative activity of these reproductive organs are not affected by the treatment with I3C. This result is in line with several clinical trials assessing the therapeutic efficacy of I3C in the treatment of cervical dysplasia as well as breast and prostate cancer [39,61,62,63]. They showed that supplementation with I3C up to 400 mg/day only elicits few, if any, adverse effects [64]. Nonetheless, we are aware that for a broad clinical application of I3C in the prevention and therapy of endometriosis, additional studies would be necessary to characterize in detail the risk profile of I3C, particularly its long-term effects on female fertility.

There are other aspects which have not been addressed in the present study, but are of major importance for the potential future use of I3C in the management of endometriosis. In fact, in our study we only treated newly developing endometriotic lesions, which may represent an approach for the prevention of disease recurrence after complete surgical removal of endometriotic lesions in patients. Accordingly, we did not analyze the therapeutic effects of I3C on different developmental stages of endometriotic lesions as they may typically occur in patients with the initial diagnosis of endometriosis. Therefore, additional experiments starting with I3C treatment after lesion establishment should be performed. Moreover, it would be interesting to investigate whether the suppressive effects of I3C on endometriotic lesions are reversible. If so, discontinuation of I3C therapy may result in the reoccurrence of blood vessel formation inside endometriotic lesions and, thus, recovery and further growth of the ectopic endometrial tissue.

In summary, the present study demonstrates that I3C effectively inhibits the vascularization and growth of newly developing endometriotic lesions in a mouse model of endometriosis. Accordingly, this plant-derived compound may represent a promising candidate for the future treatment of this disease without inducing severe side effects. This conclusion is supported by the fact that therapeutic doses of I3C have already been proven to exhibit an acceptable side effect profile in various clinical trials.

## Figures and Tables

**Figure 1 nutrients-14-04940-f001:**
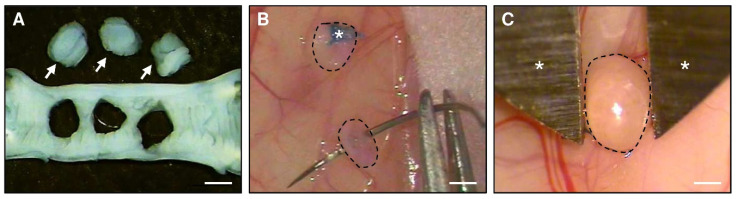
(**A**): Longitudinally opened uterine horn of a donor BALB/c mouse. Three uterine tissue samples (arrows) were excised by means of a 2-mm dermal biopsy punch. (**B**): Fixation with a Prolene suture (asterisk) of uterine tissue samples (borders marked by broken lines) to the peritoneal wall of a recipient BALB/c mouse for the induction of endometriotic lesions. (**C**): Two-dimensional measurement of the size of an endometriotic lesion (border marked by broken line) by means of a digital caliper (asterisks) 28 days after transplantation of an uterine tissue sample to the abdominal wall of a recipient BALB/c mouse. Scale bars: (**A**,**B**) = 1.8 mm; (**C**) = 1 mm.

**Figure 2 nutrients-14-04940-f002:**
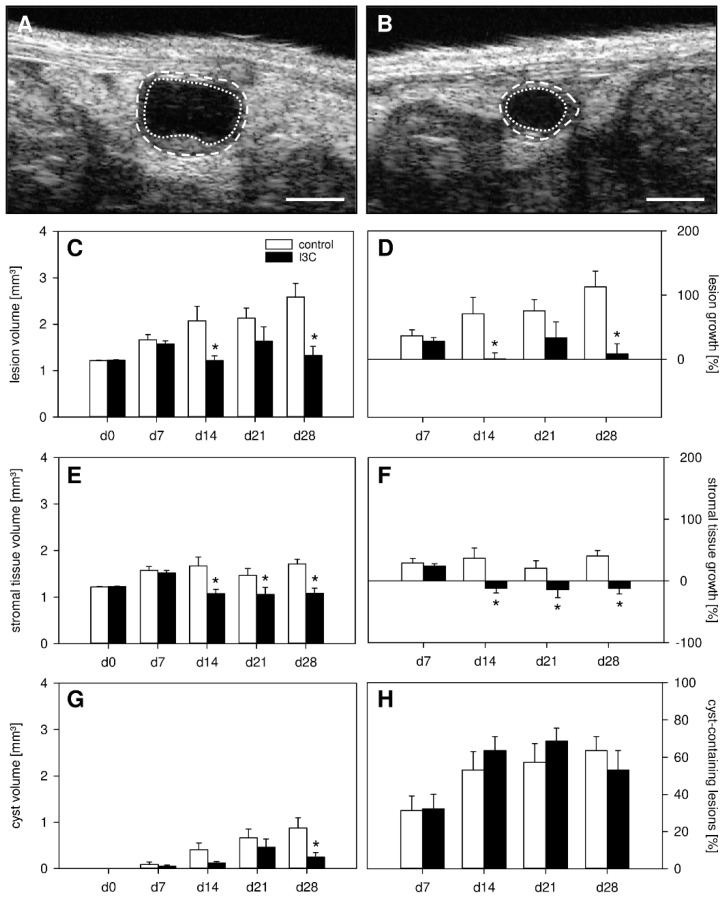
(**A**,**B**): High-resolution ultrasound imaging of endometriotic lesions (borders marked by broken lines, cyst-like dilated endometrial glands marked by dotted lines) 28 days after transplantation of uterine tissue samples to the abdominal wall of a vehicle-treated control (**A**) and an I3C-treated BALB/c mouse (**B**). Scale bars: (**A**,**B**) = 1 mm. (**C**–**H**）: Lesion volume ((**C**), mm^3^), lesion growth ((**D**), %), stromal tissue volume ((**E**), mm^3^), stromal tissue growth ((**F**), %), cyst volume ((**G**), mm^3^) and fraction of cyst-containing lesions ((**H**), %) of endometriotic lesions in vehicle-treated control (white bars; *n* = 8) and I3C-treated BALB/c mice (black bars; *n* = 8). Mean ± SEM; * *p* < 0.05 vs. control.

**Figure 3 nutrients-14-04940-f003:**
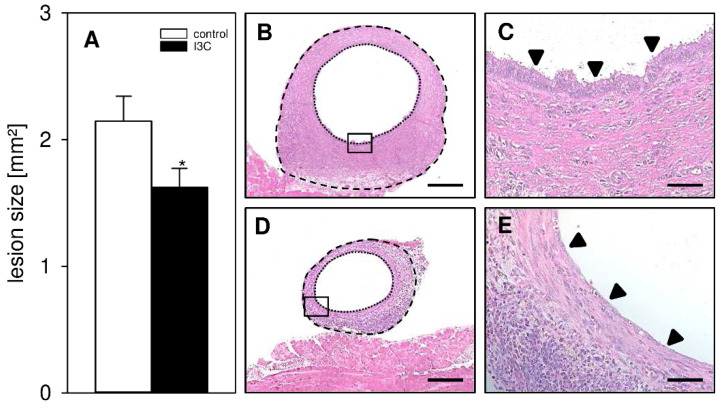
(**A**): Caliper-assessed lesion size (mm^2^) of endometriotic lesions in vehicle-treated control (white bar; *n* = 8) and I3C-treated BALB/c mice (black bar; *n* = 8). Mean ± SEM; * *p* < 0.05 vs. control. (**B**–**E**): HE-stained sections of endometriotic lesions (borders marked by broken lines, cyst-like dilated endometrial glands marked by dotted lines) 28 days after transplantation of uterine tissue samples to the abdominal wall of a vehicle-treated control (**B**,**D**) and an I3C-treated BALB/c mouse (**C**,**E**). (**D**,**E**) shows higher magnification of insert in (**B**,**C**). Scale bars: (**B**,**C**) = 200 µm; (**D**,**E**) = 50 µm.

**Figure 4 nutrients-14-04940-f004:**
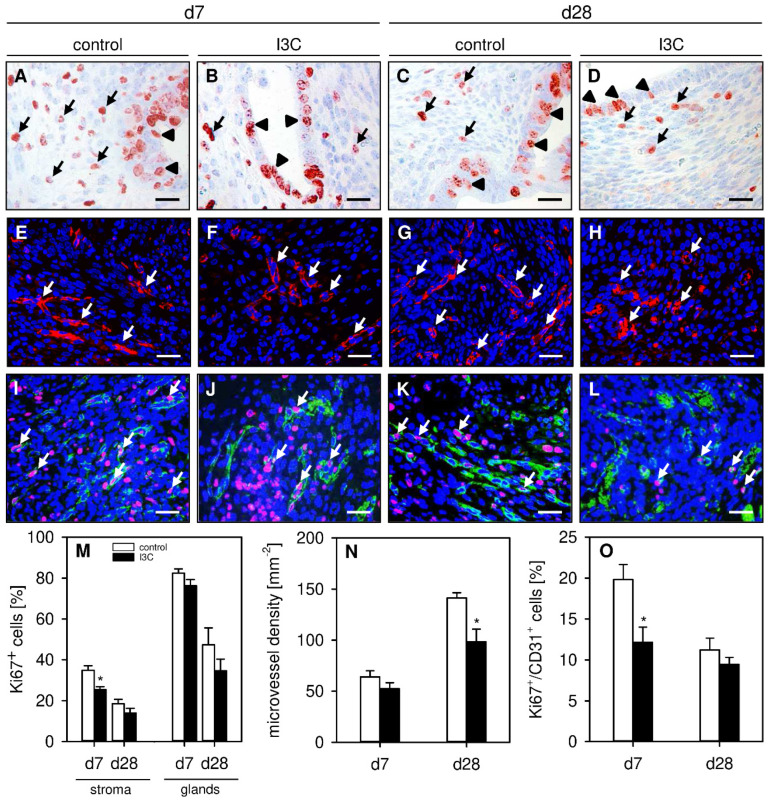
(**A**–**D**): Immunohistochemical detection of Ki67^+^ stromal cells (arrows) and glandular epithelial cells (arrowheads) on day 7 (**A**,**B**) and day 28 (**C**,**D**) after transplantation of uterine tissue samples to the abdominal wall of vehicle-treated control (**A**,**C**) and I3C-treated BALB/c mice (**B**,**D**). Scale bars: 20 µm. (**E**–**H**): Immunofluorescent detection of microvessels (arrows) on day 7 (**E**,**F**) and day 28 (**G**,**H**) after transplantation of uterine tissue samples to the abdominal wall of vehicle-treated control (**E**,**G**) and I3C-treated BALB/c mice (**F**,**H**). The immunofluorescent sections were stained with Hoechst 33342 to identify cell nuclei (blue) and an antibody against CD31 for the detection of microvessels (red). Scale bars: 30 µm. (**I**–**L**): Immunofluorescent detection of Ki67^+^/CD31^+^ endothelial cells (arrows) on day 7 (**I**,**J**) and day 28 (**K**,**L**) after transplantation of uterine tissue samples to the abdominal wall of vehicle-treated control (**I**,**K**) and I3C-treated BALB/c mice (**J**,**L**). The immunofluorescent sections were stained with Hoechst 33342 to identify cell nuclei (blue), an antibody against CD31 for the detection of endothelial cells (green) and an antibody against Ki67 for the detection of proliferating cells (red). Scale bars: 30 µm. (**M**): Ki67^+^ cells (%) within the stroma and the glands of endometriotic lesions in vehicle-treated control (white bars; *n* = 6–8) and I3C-treated BALB/c mice (black bars; *n* = 6–8). Mean ± SEM. * *p* < 0.05 vs. control. (**N**): Microvessel density (mm^−2^) of endometriotic lesions in vehicle-treated control (white bars; *n* = 6–8) and I3C-treated BALB/c mice (black bars; *n* = 6–8). Mean ± SEM. * *p* < 0.05 vs. control. (**O**): Ki67^+^/CD31^+^ cells (%) in endometriotic lesions in vehicle-treated control (white bars; *n* = 6–8) and I3C-treated BALB/c mice (black bars; *n* = 6–8). Mean ± SEM. * *p* < 0.05 vs. control.

**Figure 5 nutrients-14-04940-f005:**
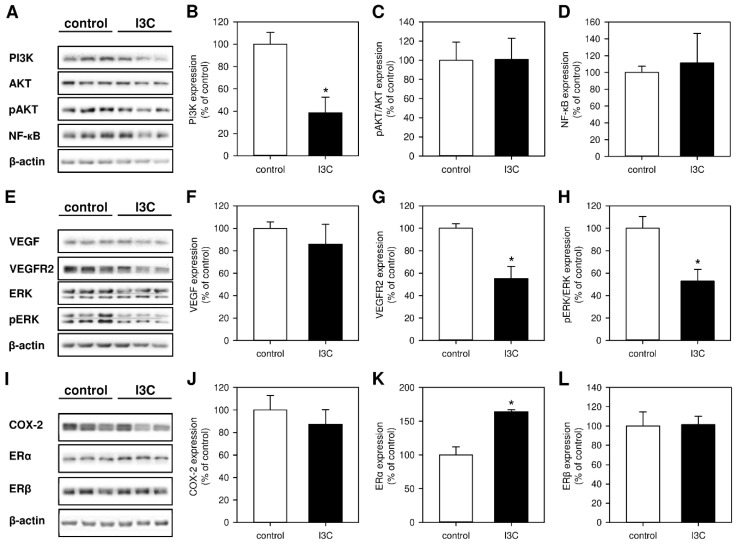
(**A**): Western blots showing the expression of PI3K, AKT, pAKT, NF-ĸB and β-actin in endometriotic lesions on day 7 after transplantation of uterine tissue samples to the abdominal wall of vehicle-treated control and I3C-treated BALB/c mice. (**B**–**D**): Expression (% of control) of PI3K (**B**), pAKT/AKT (**C**) and NF-ĸB (**D**) in endometriotic lesions of vehicle-treated control (white bars; *n* = 3) and I3C-treated BALB/c mice (black bars; *n* = 3). Mean ± SEM. * *p* < 0.05 vs. control. (**E**): Western blots showing the expression of VEGF, VEGFR2, ERK, pERK, and β-actin in endometriotic lesions on day 7 after transplantation of uterine tissue samples to the abdominal wall of vehicle-treated control and I3C-treated BALB/c mice. (**F**–**H**): Expression (% of control) of VEGF (**F**), VEGFR2 (**G**) and pERK/ERK (**H**) in endometriotic lesions of vehicle-treated control (white bars; *n* = 3) and I3C-treated BALB/c mice (black bars; *n* = 3). Mean ± SEM. * *p* < 0.05 vs. control. (**I**): Western blots showing the expression of COX-2, ERα, ERβ and β-actin in endometriotic lesions on day 7 after transplantation of uterine tissue samples to the abdominal wall of vehicle-treated control and I3C-treated BALB/c mice. (**J**–**L**): Expression (% of control) of COX-2 (**J**), ERα (**K**) and ERβ (**L**) in endometriotic lesions of vehicle-treated control (white bars; *n* = 3) and I3C-treated BALB/c mice (black bars; *n* = 3). Mean ± SEM. * *p* < 0.05 vs. control.

**Figure 6 nutrients-14-04940-f006:**
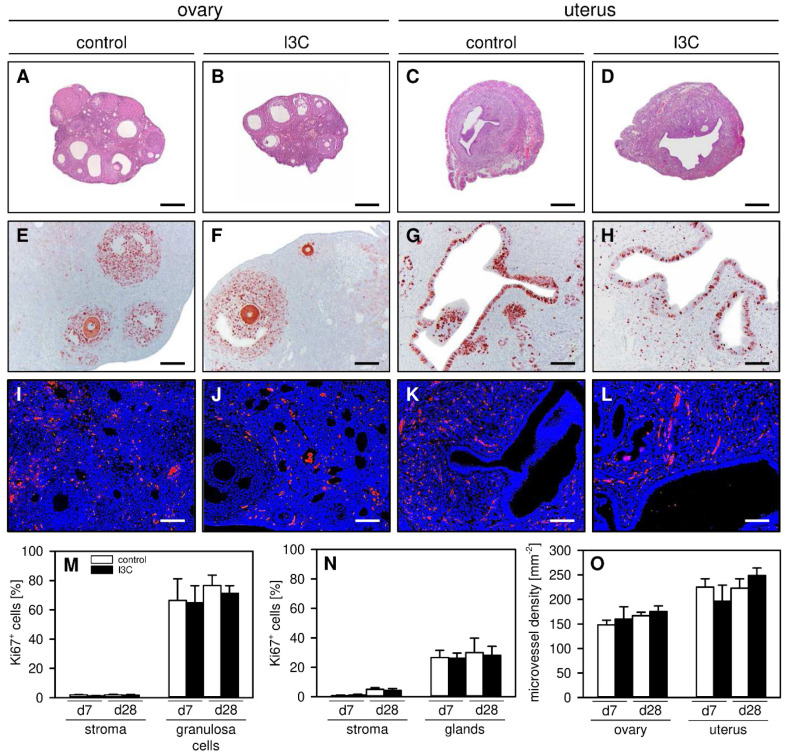
(**A**–**D**): HE-stained sections of the ovaries (**A**,**B**) and uterine horns (**C**,**D**) on day 28 after transplantation of uterine tissue samples to the abdominal wall of vehicle-treated control (**A**,**C**) and I3C-treated BALB/c mice (**B**,**D**). Scale bars: 270 µm. (**E**–**H**): Immunohistochemical detection of Ki67^+^ cells (red) in the ovaries (**E**,**F**) and uterine horns (**G**,**H**) on day 28 after transplantation of uterine tissue samples to the abdominal wall of vehicle-treated control (**E**,**G**) and I3C-treated BALB/c mice (**F**,**H**). Scale bars: (**E**–**H**) = 80 µm; (**I**–**L**) = 40 µm. (**I**–**L**): Immunofluorescent detection of microvessels in the ovaries (**I**,**J**) and uterine horns (**K**,**L**) on day 28 after transplantation of uterine tissue samples to the abdominal wall of vehicle-treated control (**I**,**K**) and I3C-treated BALB/c mice (**J**,**L**). The immunofluorescent sections were stained with Hoechst 33342 to identify cell nuclei (blue) and an antibody against CD31 for the detection of microvessels (red). (**M**,**N**): Ki67^+^ cells (%) within the stroma and the follicles of the ovaries (**M**) as well as the stroma and glands of the uterine horns (**N**) in vehicle-treated control (white bars; *n* = 5–8) and I3C-treated BALB/c mice (black bars; *n* = 6–8). Mean ± SEM. (**O**): Microvessel density (mm^−2^) of the ovaries and uterine horns in vehicle-treated control (white bars; *n* = 5–8) and I3C-treated BALB/c mice (black bars; *n* = 6–8). Mean ± SEM.

## Data Availability

The data that support the findings of this study are available from the corresponding author upon reasonable request.

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
