# Peer review of "Indole-3-Carbinol Inhibits the Growth of Endometriotic Lesions by Suppression of Microvascular Network Formation"

_nutrients, 2022, doi:10.3390/nu14224940_

Round 1
Reviewer 1 Report
The authors conducted a well designed study. However, to suggest that I3C could be used to prevent/treat human disease, some more parameters need to be tested.
This should be mentioned before summary in "future directions" section.
For example, in human disease, the disease cannot be identified on day 0 as in mouse models. Therefore, experiments with drug starting later in the disease progression should be tested. Experiments checking whether the effect of I3C is reversible also need to be tested. Does discontinuing I3C cause increased microvascular density? These questions must be answered before suggesting the use of I3C in humans.
These suggestions must be mentioned in the future directions section of the article, before the summary.
Author Response
Review of the manuscript nutrients-1997911 by Rudzitis-Auth et al.
Reply to the comments of reviewer 1
We appreciate the fair and constructive comments of the reviewer. In the following, please find our point-by-point reply.
Reviewer comment: The authors conducted a well designed study. However, to suggest that I3C could be used to prevent/treat human disease, some more parameters need to be tested. This should be mentioned before summary in "future directions" section. For example, in human disease, the disease cannot be identified on day 0 as in mouse models. Therefore, experiments with drug starting later in the disease progression should be tested. Experiments checking whether the effect of I3C is reversible also need to be tested. Does discontinuing I3C cause increased microvascular density? These questions must be answered before suggesting the use of I3C in humans.
These suggestions must be mentioned in the future directions section of the article, before the summary.
Reply: According to the comment of the reviewer, we have included a novel paragraph in the discussion section of our revised manuscript, which discusses the necessary future steps to translate our findings in the clinical management of endometriosis. This paragraph reads as follows:
‘There are other aspects, which have not been addressed in the present study, but are of major importance for the potential future use of I3C in the management of endometriosis. In fact, in our study we only treated newly developing endometriotic lesions, which may represent an approach for the prevention of disease recurrence after complete surgical removal of endometriotic lesions in patients. Accordingly, we did not analyze the therapeutic effects of I3C on different developmental stages of endometriotic lesions as they may typically occur in patients with the initial diagnosis of endometriosis. Therefore, additional experiments starting with I3C treatment after lesion establishment should be performed. Moreover, it would be interesting to investigate whether the suppressive effects of I3C on endometriotic lesions are reversible. If so, discontinuation of I3C therapy may result in the reoccurrence of blood vessel formation inside endometriotic lesions and, thus, recovery and further growth of the ectopic endometrial tissue.’
(see page 8, lines 51-54; page 9, lines 1-8; marked in yellow)
Reviewer 2 Report
The authors investigated about the effects of indole-3-carbinol on endometriotic lesions. Overall the topic could be interesting but some details could be improved.
I recommend that the paper be accepted with minor revision:
a) The authors should mentioned in the abstract more details about model used.
b) In the introduction section, little previous evidence is provided about the importance of endometriosis in daily life. Incorporating comparisons with other studies would increase the strength of the paper. Please refer to doi:
10.18632/oncotarget.25823; 10.3389/fgwh.2022.856316; 10.1038/s41598-020-79578-3; 10.3390/ijms22105074. c) The authors should clarify why they use this dose of I3C? Any references?d) The authors should better emphasize the conclusions.
e) There are some minor grammar issues that should be fixed in order to aid the accessibility of the results to the reader.
Author Response
Review of the manuscript nutrients-1997911 by Rudzitis-Auth et al.
Reply to the comments of reviewer 2
We appreciate the fair and constructive comments of the reviewer. In the following, please find our point-by-point reply.
Reviewer comment: The authors should mentioned in the abstract more details about model used.
Reply: According to the comment of the reviewer, we now provide more details about the used endometriosis model in the abstract of the revised manuscript version (see page 1, lines 31-33; marked in yellow).
Reviewer comment: In the introduction section, little previous evidence is provided about the importance of endometriosis in daily life. Incorporating comparisons with other studies would increase the strength of the paper. Please refer to doi: 10.18632/oncotarget.25823; 10.3389/fgwh.2022.856316; 10.1038/s41598-020-79578-3; 10.3390/ijms22105074.
Reply: According to the comment of the reviewer, we have included the suggested references in the introduction section of our revised manuscript (see page 2, lines 6-11; page 10, lines 12-22; marked in yellow).
Reviewer comment: The authors should clarify why they use this dose of I3C? Any references?
Reply: The reasons why we used the dose of 120 mg/kg I3C has already been explained in detail in a paragraph of the discussion section of our original manuscript version. This paragraph reads as follows:
‘The animals were daily treated by peroral administration of 120 mg/kg I3C, which has previously been shown to effectively suppress murine leukemia [37]. This dose in mice corresponds to 9.7 mg/kg or 679 mg of I3C for a 70 kg person according to the dose trans-lation method by Reagan-Shaw et al. [38]. Thus, the I3C dose used in our experiments is within the dose range of 400-800 mg daily, which has already been applied in a phase-I study for the treatment of women with high risk of breast cancer [39]. Importantly, this dose range was well tolerated by the patients and did not induce severe side effects [39].’
(See page 7, lines 24-31; marked in yellow)
References:
- Lu HF, Tung WL, Yang JS, Huang FM, Lee CS, Huang YP, Liao WY, Chen YL, Chung JG. In vitro suppression of growth of murine WEHI-3 leukemia cells and in vivo promotion of phagocytosis in a leukemia mice model by indole-3-carbinol. J Agric Food Chem 2012; 60, 7634-7643.
- Reagan-Shaw S, Nihal M, Ahmad N. Dose translation from animal to human studies revisited. FASEB J 2008; 22, 659-661.
- Reed GA, Peterson KS, Smith HJ, Gray JC, Sullivan DK, Mayo MS, Crowell JA, Hurwitz A. A phase I study of in-dole-3-carbinol in women: tolerability and effects. Cancer Epidemiol Biomarkers Prev 2005; 14, 1953-1960.
Reviewer comment: The authors should better emphasize the conclusions.
Reply: According to the comment of the reviewer, we have rewritten our conclusion section, which now reads as follows:
‘In summary, the present study demonstrates that I3C effectively inhibits the vascularization and growth of newly developing endometriotic lesions in a mouse model of endometriosis. Accordingly, this plant-derived compound may represent a promising candidate for the future treatment of this disease without inducing severe side effects. This conclusion is supported by the fact that therapeutic doses of I3C have already been proven to exhibit an acceptable side effect profile in various clinical trials.’
(See page 9, lines 9-16; marked in yellow)
Reviewer comment: There are some minor grammar issues that should be fixed in order to aid the accessibility of the results to the reader.
Reply: According to the comment of the reviewer, we have carefully checked again the manuscript for grammar errors, which have been corrected in the revised version.